# Data Alignment Predicts Language Model Performance: Evidence from Controlled Experiments in Autoformalization

## Abstract

We investigate whether data alignment – the similarity between training and evaluation data – is a stronger predictor of language model performance than dataset size. Through controlled experiments, we demonstrate that alignment coefficients consistently predict downstream performance across three distinct metrics: Task2Vec embeddings ($r^2 = 0.80$–$0.96$), GZIP compression distance ($r^2 = 0.90$), and sentence embeddings ($r^2 = 0.80$). We consider two experimental settings: (1) pre-training on domain-specific corpora (PubMed, USPTO) and evaluating cross-domain performance, and (2) fine-tuning on autoformalization datasets with varying alignment to formal verification tasks. Our results show strong negative correlations between alignment and perplexity across both settings, with highly aligned small datasets (1.4k tokens) outperforming larger misaligned datasets (4.1k tokens) by 53% in perplexity reduction. These findings provide quantitative evidence that strategic data selection based on alignment can be more effective than simply scaling dataset size, offering practical guidance for efficient model training in specialized domains.

## 1 Introduction

Research within the domain of Large Language Models (LLMs) has historically placed an emphasis on the size of datasets used for pre-training, claiming it is one of the primary determinants of LLM performance (Chowdhery et al., 2022; Nostalgebraist, 2022; OpenAI, 2023; Google, 2023b). Empirical evidence demonstrates this trend, as models trained on large datasets exhibit superior performance. Notably, GPT-4, with its conjectured 1 petabyte dataset, markedly surpasses GPT-3—which is trained on a comparatively modest 45 terabytes—in terms of response quality and contextual accuracy (OpenAI, 2023). However, emerging research indicates that other dimensions, such as dataset diversity, play a crucial role in the efficacy of LLMs, with high-performing models often arising from datasets with high diversity coefficients (Lee et al., 2023).

Current discourse predominantly highlights the scale of a dataset as a pivotal factor in its capacity to effectively pre-train or fine-tune a model, with emphasis frequently placed on quantitative metrics—specifically, the sheer size of the dataset (Lee et al., 2023). This investigation, however, seeks to shift this paradigm to consider qualitative assessments, notably the alignment of datasets with the specific evaluation tasks. Building upon methodologies established in previous studies for quantifying dataset alignment, our research aims to examine the role of data quality in the pre-training and fine-tuning process, verifying the hypothesis that increased data alignment could significantly improve LLM performance. This paradigm challenges the emphasis on dataset size, suggesting an alternative approach to dataset importance and optimization in the context of LLM training – i.e., select the most aligned data to your target task. We explore this via Autoformalization.

Autoformalization is defined as the transformation of concepts in natural language to formalized, structured language like mathematical proofs or code. The creation of a proficient Autoformalization tool would not only drastically reduce the substantial costs associated with manual formalization efforts but could also serve as a bridge linking the automated theorem verification and computational algebra with the extensive body of mathematical knowledge predominantly recorded in natural language. Moreover, the capacity for Autoformalization underscores a machine's adeptness at navigat-

ing the subtleties of human language and the precision required by formal linguistic systems (Wu et al., 2022).

We employ a comprehensive evaluation by comparing the performance of fine-tuned LLMs on *quantitatively* aligned data sets against those calibrated primarily for scale. We engage a broad spectrum of Autoformalization tasks across different domains and complexities, ensuring the thoroughness and robustness of our results.

## 2 METHODS

Our experiment is designed to explore the hypothesis that there exists a negative correlation between the alignment score of a dataset with a benchmark and the perplexity score (see Appendix C) of a Large Language Model (LLM) when either pre-trained or fine-tuned on this dataset and evaluated against said benchmark. The crux of our investigation lies in the assertion that a dataset closely aligned with the benchmark will facilitate the LLM's learning process, thereby enhancing its performance as evidenced by lower perplexity scores.

### 2.1 CONCEPTUAL FRAMEWORK

The alignment score is a critical metric in our analysis, offering insight into the degree of congruence between a dataset and the chosen benchmark for evaluating downstream performance, such as Autoformalization. We posit that an LLM trained on a dataset that mirrors the characteristics of the benchmark will demonstrate superior performance. This performance is quantitatively measured using the perplexity score. For example, in the fine-tuning setting we measure model perplexity on the debug1AF benchmark, where lower scores denote higher model accuracy and effectiveness.

### 2.2 DATASET ALIGNMENT QUANTIFICATION

To quantify dataset alignment, we employ the Task2Vec Alignment Coefficient, which facilitates a rigorous comparative assessment of dataset similarity (Lee et al., 2023).

The alignment coefficient between two datasets, $D_1$ and $D_2$, is calculated as:

$$\hat{align}(D_1, D_2) = 1 - \mathbb{E}_{B_1 \sim D_1, B_2 \sim D_2}[d(\hat{f}(B_1), \hat{f}(B_2))] \tag{1}$$

where $\mathbb{E}$ denotes the expectation over batches $B_1$ and $B_2$ sampled from datasets $D_1$ and $D_2$, respectively, and $d(\hat{f}(B_1), \hat{f}(B_2))$ represents the distance between the embeddings of these batches. Unless otherwise specified, $d$ is defined as cosine distance and is derived through $\hat{f}$ which is the Task2Vec batch-embedding computed with a fixed probe network (GPT-2 in our experiments) by estimating the diagonal Fisher information of the probe's parameters on $B$ and flattening it to a vector Achille et al. (2019).

For the purposes of our experimental framework, we consider the alignment of the entire dataset rather than focusing solely on specific subsets. We assume that the alignment properties of a dataset subset are reflective of the dataset as a whole. Consequently, our alignment evaluations are predicated on the comprehensive dataset, offering a holistic view of dataset congruence and its impact on model performance.

## 3 EXPERIMENTS & RESULTS

### 3.1 EFFECTS OF DATA ALIGNMENT BETWEEN PRE-TRAINING AND EVALUATION DATA

#### 3.1.1 EXPERIMENTAL SETUP AND MOTIVATION

To evaluate the effect of data alignment between pre-training data and downstream task, we pre-train 51M parameter GPT-2 models (Radford et al., 2019) for 1.31B tokens on one of three datasets: PubMed Abstracts, a dataset of medicine-related abstracts; USPTO Backgrounds, a dataset of patent application background sections; and a dataset produced by concatenating USPTO and PubMed Abs.

By controlling for all training hyperparameters aside from the pretraining dataset, we minimize the effect of confounding variables on the relationship between data alignment and downstream performance. We proceed to evaluate these pre-trained models on a variety of evaluation datasets, which vary in terms of their similarity to the three pre-training datasets, both empirically in terms of the alignment coefficient and qualitatively based on the topic and structure of text within each dataset. By evaluating language modeling cross-entropy loss for a given pre-trained model on a given evaluation dataset, we directly test the importance of pre-training data alignment with the model's downstream task in order to illustrate the relationship between alignment and downstream performance.

### 3.1.2 Pre-training experiment results

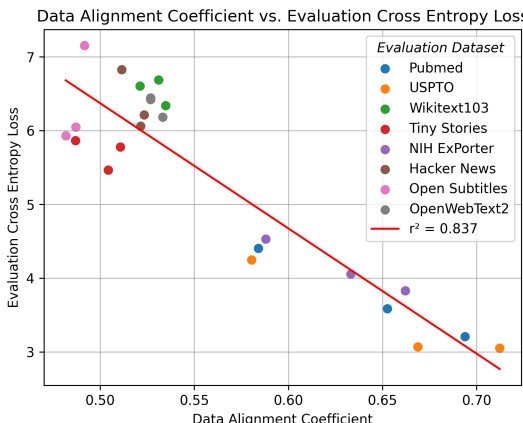

Figure 1: The data alignment coefficient demonstrates a strong relationship with model performance (cross-entropy loss) on various evaluation datasets ($r^2 = 0.8$). The data alignment coefficient is computed between a model's pre-training dataset (PubMed Abs., USPTO, or PubMed Abs. + USPTO) and a single evaluation dataset (represented by a unique color).

Figure 1 demonstrates that there is a moderate–strong relationship between the alignment coefficient (between pre-train data and evaluation data) and model performance (cross-entropy loss) pooled across various evaluation datasets ($r^2 = 0.8$). As expected, when datasets share similarities in topic and structure, the alignment coefficient is higher.

These results demonstrate that the alignment between pre-training corpora and evaluation data is a significant driver of model performance. For instance, when considering the extremes of alignment and lack thereof, the most aligned train-evaluation data (USPTO train with USPTO validation data) produces approximately 2.9 lower absolute CE loss compared to the least aligned train-evaluation data (PubMed Abs. train with Open Subtitles validation data). Furthermore, an important aspect of model performance with respect to its alignment coefficient is that the relationship between performance and alignment demonstrates a strong, predictable downward trend, more rigorously characterizing the relationship between alignment and downstream performance than a qualitative intuition of superior performance with greater alignment.

### 3.2 Effects of Data Alignment Between Fine-Tuning and Evaluation Data

### 3.2.1 Experimental Setup and Motivation

In order to test whether an LLM will be better able to perform AF when fine-tuned on a dataset that is closely aligned to the AF benchmark, we must fine-tune LLMs on datasets of differing alignment to the benchmark. This allows us to observe a relationship between alignment and perplexity loss.

We chose to run our experiment on the following datasets specifically to introduce a controlled range of alignment values in our results. To do so, we selected datasets that represented diversity in both domain relevance and data structure. While not exhaustive, these datasets represent three

major regimes in LLM training and evaluation: natural language prose, formal mathematics, and multi-language code.

1. AF Dataset (AF): A dataset consisting of informal statements and their formal counterparts in Isabelle designed for training LLMs to perform Autoformalization. We use its test set as a benchmark of LLM performance on statement Autoformalization. Thus, we also believe it will result in the lowest perplexity among the proof datasets when used to train an LLM for AF (Miranda, 2021).

2. Destructed AF Dataset (AF-split): This dataset is composed of the AF Dataset's formal and informal statements but the two are split into different lines so that the LLM trains on data that does not explicitly indicate a relationship between the two; we expect this to still obtain a relatively low perplexity score given its high alignment.

3. The Stack Smol Python Docstrings dataset (Docstring): A dataset consisting of concise function headers written in informal language and their implementations in Python; we use it to assess how well coding datasets can fine-tune for Autoformalization (Bird, 2023a).

4. The Stack Dedup Python Docstrings 1.0 percent unified dataset (Docstring 2): A dataset consisting of function headers written in informal language and their implementations in Python; given its nature we anticipate it scoring among the lowest of perplexity scores against the Docstring benchmark (Bird, 2023b).

5. C4-EN-10K Dataset (C4): A ten-thousand-entry subset of a database composed of text pulled from Common Crawl (an internet archive) meant for pre-training for general English language modeling. Given its entries are all informal statements not related to mathematics, we predict a high perplexity score in performing AF (Raffel et al., 2019).

6. wikitext-2-raw-v1 Dataset (Wikitext): A subset of the Wikitext dataset; Wikitext is a dataset composed of text taken from Wikipedia pages that met the score guidelines to qualify as either a 'good' or 'featured' article; given its nature and lack of relevance to AF, we expect a high perplexity score (Merity et al., 2016).

7. minif2f-lean4 Dataset (LeanDojo4): A subset of the miniF2F dataset which is comprised of math exercise statements and their formal counterparts in Lean; given that it is in a different formal language, we expect a mid-range perplexity score (Zheng et al., 2021).

8. Proofnet Dataset (Proofnet): This dataset is comprised of statements taken from undergraduate math courses and their formal counterparts in Lean; given their similarities, we expect LeanDojo4 and Proofnet to score similarly in perplexity (Azerbayev et al., 2023).

9. HumanEvalPack: This dataset consists of a prompt describing a function and implementations of the function in Python, JavaScript, Java, Go, C++, and Rust as well as buggy solutions to serve as bad examples. We expect it to obtain a mid-range score against the Docstring benchmark (Muennighoff et al., 2023).

For each of these datasets, we needed to separate them into proof datasets and code datasets and preprocess the data accordingly. Figure 3 visualizes our method.

### 3.2.2 ADDITIONAL CORRELATION RESULTS

Table 1: All datasets and their corresponding number of tokens.

| Dataset | Number of Tokens |
|---|---|
| AF | 4092 |
| C4 | 4096 |
| Wikitext | 4186 |
| Proofnet | 4032 |
| LeanDojo4 | 4186 |
| ProofPile | 4096 |
| Docstring-Python | 4116 |
| Humanevalpack | 4004 |
| Docstring-Python-2 | 3790 |

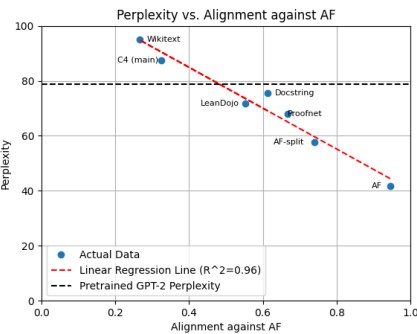
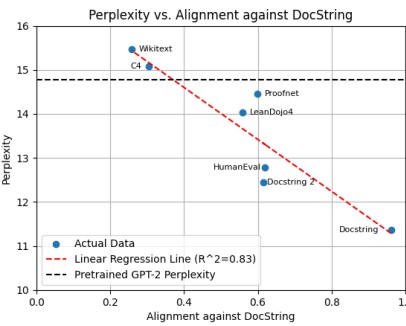

Figure 2: Alignment scores plotted against perplexity suggest a linear negative correlation and mirror our expected findings described in the evaluation design. left plot shows negative correlation of alignment and test perplexity for autoformalization. right plot shows negative correlation of alignment and test perplexity for docstring to code.

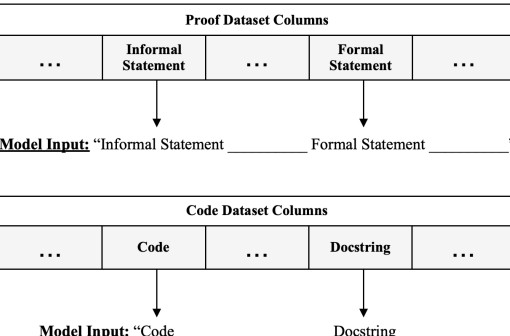

Figure 3: Data preprocessing workflow for separating code and proof datasets.

### 3.3 ANALYSIS OF RESULTS

Introduction of Data Alignment in LLM Training: A novel approach that integrates data alignment as a key factor in the training of Large Language Models, leading to improved model performance.

1. Empirical Evidence of Alignment Impact: Through systematic experimentation, the paper provides empirical evidence that higher alignment between training data and the target domain leads to a decrease in perplexity scores, indicative of enhanced model accuracy.

2. Analysis Across Multiple Datasets: The study conducts a comprehensive analysis across a variety of datasets, establishing the consistency of the negative correlation between data alignment and perplexity across both proof and code datasets.

3. Demonstration of a High $r^2$ Correlation Value: Our experiments demonstrate a robust negative correlation between data alignment and model perplexity, with a high $r^2$ value of 0.96 for proof datasets and 0.83 for code datasets when evaluated on Autoformalization tasks, and an $r^2$ of 0.8 for a pre-training setting with a variety of training and evaluation datasets, indicating a strong predictive relationship.

4. Identification of Limitations and Future Research Avenues: The paper discusses the limitations of the current study due to hardware constraints and sets the stage for future research to explore the comparative impact of dataset size versus alignment.

#### 3.3.1 PROOF DATASET RESULTS

For each dataset we calculated the alignment scores using Task2Vec Alignment Coefficient as depicted in Table 2. Our final perplexity scores for each of our models trained can be found in Table 3. This can be difficult to visualize, so we plotted our results as shown in Figure 2.

Our results are significant in validating our initial thesis that a highly aligned data is capable of producing an LLM that performs better than one that was trained on a dataset with lower alignment.

We found that an untuned, standard gpt-2 LLM received a perplexity score of 78.7413. However, after finetuning it on AF it received the best perplexity score in our results: 41.8261. This further bolsters our claim as AF-AF also had the greatest alignment (approximately 0.945) and the best performance as well.

The proofnet dataset did not perform as well as AF fine tuned, with a perplexity score of 67.8906 and alignment of 0.67. However, this is expected based on our thesis as we see that a drop in alignment contributes to an increase in perplexity score for the model.

The C4 dataset has a much lower score in alignment (approximately 0.32) compared to AF (approximately 0.95). Judging by this metric alone, we would expect to see a higher perplexity than gpt-2 finetuned by the debug1AF dataset based on our thesis. When fine-tuning gpt-2 on a subset of C4, this proved to be the case as the perplexity score is 87.4636, about 11% higher than Standard gpt-2 and 110% higher than AF fine tuned.

Furthermore, the dataset with the worst alignment, Wikitext, with a alignment coefficient of approximately 0.27 performed poorly: the perplexity score of 94.9470 is clearly the worst amongst our datasets. This backs up our initial claim.

Ultimately, there is a clear negative correlation between the alignment coefficient and perplexity, as depicted in the graphs above: we observe an $r^2$ value of approximately 0.987 in the left-hand plot in Figure 2, suggesting a strong linear fit. The slope of the fitted linear function is approximately -74.4, demonstrating that a 0.1 increase in the alignment coefficient correlates with a decrease in perplexity of approximately 7.4. Importantly, the negativity of the slope demonstrates the negative correlation between alignment and perplexity.

### 3.3.2 CODE DATASET RESULTS

For our code dataset, we found again a strong negative correlation between alignment score and perplexity loss. Immediately we see that perplexity scores are much lower for the code datasets than the proof datasets. This is probably due to the fact that GPT-2 knows how to generate code quite well based solely on pre-training, as it has been pre-trained on a large, diverse web-based pre-training corpus (Radford et al., 2019). As a result, fine-tuning further on code produces even greater results. Standard GPT-2 has a baseline perplexity score of 14.8, which is quite good and is indicated by the dotted gray line in Figure 2.

We see that the baseline Docstring-Docstring has an alignment score close to 1 (0.96) and as a result has the lowest perplexity score (11.4), performing the best out of all our fine-tuned models. Moreover, The model that performs the worst also has the lowest alignment of 0.26, Docstring-Wikitext.

As with the proof datasets, we see a negative correlation between alignment and perplexity, with an $r^2$ value of 0.85. While this is not as high as 0.987 as we observed in the proof dataset, this is still a strong correlation and further reinforces our thesis.

### 3.4 IMPACT OF DATA ALIGNMENT VERSUS DATASET SIZE ON LLM PERFORMANCE

This experiment provides an illustrative contrast between the impact of data alignment with the downstream task and that of the size of the dataset used for fine-tuning. We hypothesized that a smaller, highly aligned dataset would lead to better LLM performance on the downstream task of Autoformalization, as measured by perplexity loss, compared to a larger but less aligned dataset.

Two datasets were used for fine-tuning a pre-trained GPT-2 model:

1. A small dataset, extracted directly from the debug1AF benchmark, comprising approximately 1.4k tokens. This dataset was expected to have high alignment (close to 1) with the Autoformalization task, given its direct sampling from the task's benchmark.

2. A larger, mixed dataset designed to have a lower alignment score of 0.54 with the debug1AF benchmark. The dataset size was significantly larger than the first (approximately 4100 tokens), intended to test the effect of dataset size versus alignment.

Both models were fine-tuned under identical conditions, barring the training dataset, and evaluated on the debug1AF benchmark to measure performance through perplexity loss.

The results of the fine-tuning experiment support our hypothesis regarding the importance of data alignment. The model fine-tuned on the smaller, highly aligned dataset achieved a perplexity loss of 32.42 on the debug1AF benchmark. In contrast, the model fine-tuned with the larger, less aligned dataset exhibited a higher perplexity loss of 69.06, indicating lower performance on the Autoformalization task.

These results highlight the importance of data alignment over dataset size in LLM fine-tuning for tasks like Autoformalization. A smaller, highly aligned dataset yielded better performance than a larger, less aligned one. This anecdotally supports our hypothesis that prioritizing data quality and alignment with the task at hand will result in a higher-performing model than one that is trained on a dataset selected for sheer quantity. Consequently, we recommend a focused approach to dataset selection and preparation, prioritizing alignment to improve LLM performance on specific downstream tasks. Future experimentation is needed to isolate the impact of scale and alignment on downstream performance independently and to quantify the relative impact of each.

### 3.5 METRIC-AGNOSTIC ALIGNMENT–LOSS TREND

Replacing Task2Vec with two alternatives—**GZIP-Align** (compression distance) and **SBERT-Align** (embedding cosine)—keeps the same linear drop: higher alignment $\rightarrow$ lower loss (Fig. 4). On ProofNet, GZIP reaches $r^2 = 0.90$ and SBERT $r^2 = 0.80$, close to T2V's $r^2 = 0.88$, suggesting that the alignment–performance link is metric-agnostic. Future experimentation is needed to establish if this trend holds regardless of the dataset being tested.

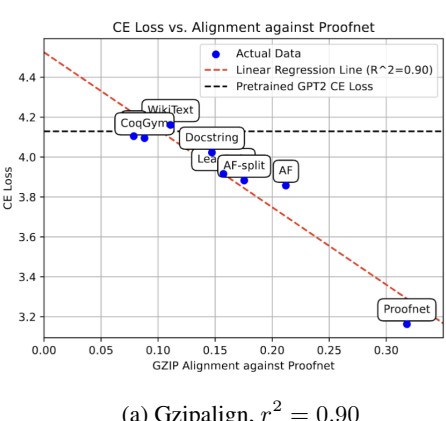
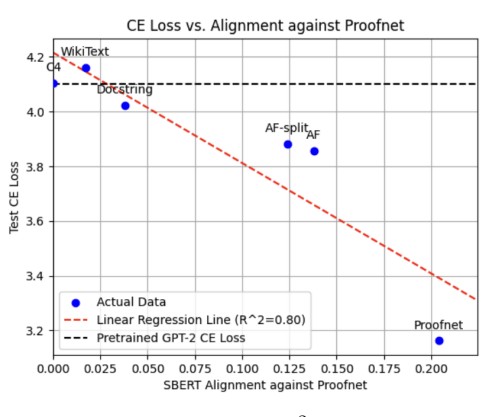

(a) Gzipalign, $r^2 = 0.90$          (b) SBERT-Align, $r^2 = 0.80$

Figure 4: Alignment–loss correlation on the ProofNet dataset persists across metrics: compression-based (left) and embedding-based (right) scores both show the same negative trend observed with Task2Vec.

ACKNOWLEDGMENTS

Use unnumbered third level headings for the acknowledgments. All acknowledgments, including those to funding agencies, go at the end of the paper.

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

## A    APPENDIX

## B    DETAILED PRE-TRAINING ALIGNMENT EXPERIMENT RESULTS

In Table 4, we detail the specific alignment coefficient values between (pre-)training and evaluation data with 95% confidence intervals. Once again, we observe that increased alignment coefficients between train and evaluation data show a strong trend of leading to lower evaluation loss.

## C    PERPLEXITY CALCULATION

Perplexity serves as a measure of a model's prediction accuracy, with lower values indicating better performance. It is calculated using the following formula:

$$PPL(X) = \exp\left\{-\frac{1}{t}\sum_i \log p_\theta(x_i|x_{<i})\right\} \tag{2}$$

where $PPL(X)$ denotes the perplexity of sequence $X$, $t$ is the total number of tokens in $X$, $x_i$ is the $i$th token, $x_{<i}$ represents all tokens preceding $x_i$, and $\log p_\theta(x_i|x_{<i})$ is the log-likelihood of token $x_i$ given its preceding context as predicted by the model parameters $\theta$.

Table 2: Alignment scores of proof datasets on the AF benchmark.

| Datasets | Alignment score |
|---|---|
| AF-AF | 0.9452813267707825 |
| AFSplit-AF | 0.7397596240043640 |
| AF-Proofnet | 0.6674373149871826 |
| AF-Docstring | 0.6128289103507996 |
| AF-LeanDojo4 | 0.5514505505561829 |
| AF-C4 | 0.3249419331550598 |
| AF-Wikitext | 0.26609545946121216 |

Table 3: Perplexity loss for models fine-tuned on proof datasets.

| Model | Perplexity |
|---|---|
| Standard GPT-2 | 78.7413 |
| AF fine-tuned | 41.8261 |
| Proofnet fine-tuned | 67.8906 |
| LeanDojo4 fine-tuned | 71.8377 |
| C4 fine-tuned | 87.4636 |
| Wikitext fine-tuned | 94.9470 |
| Docstring fine-tuned | 75.4504 |

# D   DATA ALIGNMENT RESULTS

Table 4: **The data alignment coefficient appears to capture an intuitive notion of data similarity, since it finds training data that shares similar semantics and structure as the validation data as most aligned.** In particular, PubMed Abs. (train) and NIH Exporter, which share the semantics of health-related research and the structure of being research writing, are found to be more aligned than USPTO (patent application backgrounds). Similarly, USPTO + PubMed Abs. (train) is more aligned to USPTO (validation) than PubMed Abs. (train), but less aligned to USPTO (validation) than USPTO (train), as expected. Each cell indicates the alignment coefficient between the given pre-training dataset (row label) and evaluation dataset (column label).

| Pre-training dataset | USPTO (validation) | PubMed Abs. (validation) | OpenWebText2 |
| --- | --- | --- | --- |
| USPTO | $0.7123 \pm 0.001717$ | $0.5840 \pm 0.001389$ | $0.5267 \pm 0.001377$ |
| PubMed Abs. | $0.5805 \pm 0.001396$ | $0.6939 \pm 0.001697$ | $0.5268 \pm 0.001367$ |
| USPTO + PubMed Abs. | $0.6687 \pm 0.001602$ | $0.6526 \pm 0.001513$ | $0.5332 \pm 0.001390$ |

| Pre-training dataset | NIH Exporter | Hacker News | Open Subtitles |
| --- | --- | --- | --- |
| USPTO | $0.5879 \pm 0.001388$ | $0.5234 \pm 0.001275$ | $0.4917 \pm 0.001162$ |
| PubMed Abs. | $0.6622 \pm 0.001569$ | $0.5114 \pm 0.001300$ | $0.4817 \pm 0.001145$ |
| USPTO + PubMed Abs. | $0.6331 \pm 0.001452$ | $0.5215 \pm 0.001272$ | $0.4871 \pm 0.001123$ |

| Pre-training dataset | Wikitext-103 | Tiny Stories |
| --- | --- | --- |
| USPTO | $0.5311 \pm 0.001303$ | $0.5107 \pm 0.001203$ |
| PubMed Abs. | $0.5212 \pm 0.001200$ | $0.4868 \pm 0.001167$ |
| USPTO + PubMed Abs. | $0.5347 \pm 0.001290$ | $0.5042 \pm 0.001169$ |

# E   EXPERIMENT TO VERIFY THAT EACH SUBSET WILL HAVE A SIMILAR PERPLEXITY LOSS TO THAT OF THE ENTIRE DATASET

We have examined the perplexity loss of one subset of the dataset on which we have trained on rather than the perplexity score of the entire dataset. However, we conducted an experiment to show that these two values are comparable. We have kept the token sizes around 4000 tokens as such:

Table 5: Subsets and their corresponding number of tokens.

| Subset | Number of tokens |
| --- | --- |
| C4 Subset Original | 4096 |
| C4 Subset 1 | 4032 |
| C4 Subset 2 | 4080 |
| C4 Subset 3 | 3990 |

Then, we calculated the perplexity score for each of these subsets exactly as outlined in the *Evaluation* section. Here are the results:

Table 6: Perplexity scores for C4 fine-tuned model.

| C4 subset | Perplexity |
| --- | --- |
| Original subset | 87.4636 |
| Subset 1 | 84.4889 |
| Subset 2 | 85.9207 |
| Subset 3 | 87.4829 |

Here is the graph of all the subsets of C4 along with our original proof dataset fine tuned models:

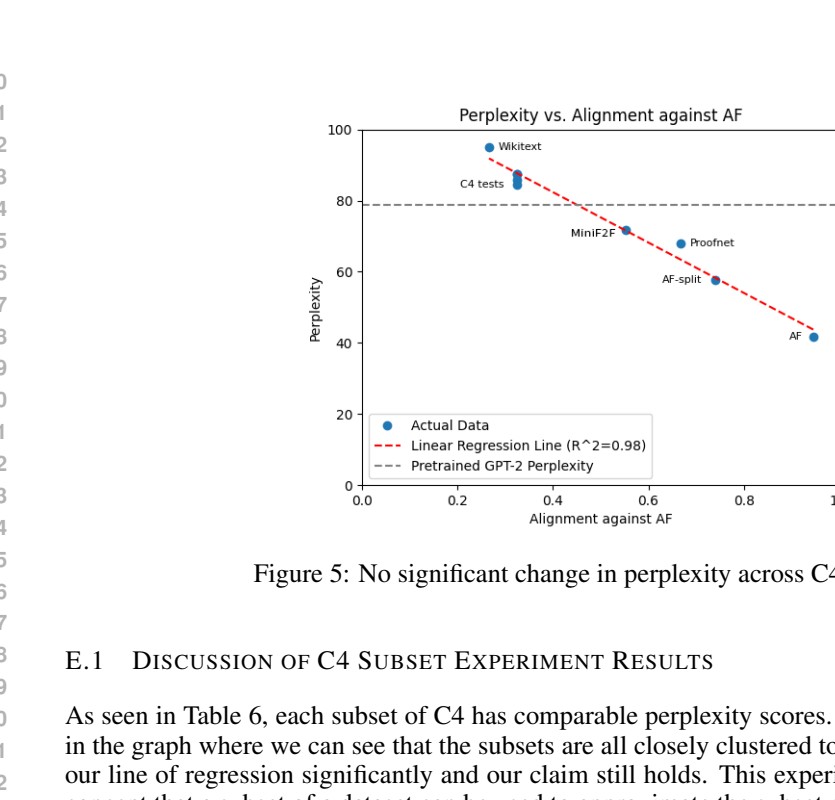

Figure 5: No significant change in perplexity across C4 subsets.

### E.1 DISCUSSION OF C4 SUBSET EXPERIMENT RESULTS

As seen in Table 6, each subset of C4 has comparable perplexity scores. This is further highlighted in the graph where we can see that the subsets are all closely clustered together; this does not affect our line of regression significantly and our claim still holds. This experiment serves as a proof-of-concept that a subset of a dataset can be used to approximate the subset of the entire dataset.

## F EXPERIMENT ON SPLITTING FORMAL AND INFORMAL STATEMENTS IN THE TRAINING PROCESS:

So far we have pre-processed our data as depicted in Figure 3, where each input contains a formal and informal statement (proof dataset) or code and docstring (code dataset). However, we conducted an experiment to observe if inputting formal and informal statements as separate inputs and training on that would produce better results. Figure 6 depicts what this would look like.

We compared the results of AF and AF-Split as follows. We first standardized the number of tokens to 4000 as seen in table 7.

Then, we calculated the alignment as shown in Table 2.

Table 7: AF and AF split tokens.

| Subset | Number of tokens |
|---|---|
| AF Original | 4092 |
| AF Split | 3960 |

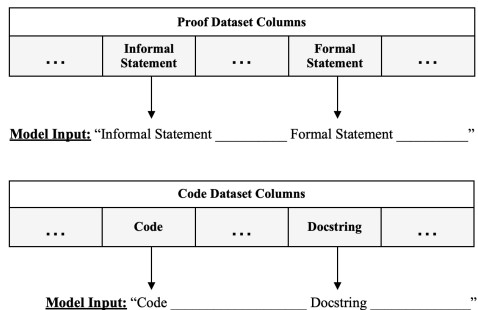

Figure 6: Data preprocessing visualization.

Finally, we fine-tuned the model on AF-Split and compared the perplexity loss to AF; this is depicted in Table 8.

Table 8: Perplexity loss scores for AF and AF split.

| Model | Perplexity |
|---|---|
| AF fine-tuned | 41.8261 |
| AF split fine-tuned | 57.8004 |

### F.1 DISCUSSION OF AF-SPLIT EXPERIMENT OUTCOMES

The investigation revealed a discernible reduction in alignment for the AF-Split dataset by approximately 21.7 percent, which constitutes a moderate deviation. Furthermore, there was a notable increase in perplexity loss for AF-Split, approximately 38.2 percent underscoring a significant impact. These findings suggest that models are more adept at Autoformalization tasks when trained on datasets that present related information cohesively, rather than on datasets where related content is disjointed. Specifically, models excel in Autoformalization when they can discern the intrinsic connection between an informal and a formal statement, as exemplified in the format "Informal Statement ____ Formal Statement ____ ," implying an inherent correlation. Conversely, when such relational cues are absent, as in the case of AF-Split where informal and formal statements are segregated, model performance in Autoformalization tasks diminishes.

### F.2 RELATED WORK (CONT.)

The article Google (2023a) "PaLM 2 Technical Report" by Google discusses the development and performance of PaLM 2. The study showcases PaLM 2's versatility but also emphasizes the role of architectural enhancements and diverse model objectives in achieving superior results. The inclusion of a diverse data mixture, even incorporating a small amount of translation pairs, results in performance comparable to dedicated translation services, a statement which supports our belief that data quality can be a critical factor in determining how well a dataset can train an LLM. This sentiment is also expressed in the article "Model Performance Scaling with Multiple Data Sources" by Tatsunori Hashimoto.Hashimoto (2021) It discusses the challenges of training ML models using data from various sources that vary in quality and cost. Hashimoto proposes a parametric model to approximate generalization error, which is more accurate for various models compared to existing linear approximations. The work represents a step toward better understanding model performance under varying data conditions and questions whether the approach can scale to more extreme scenarios or larger numbers of data sources in future research.

"Random Network Distillation as a Diversity Metric for both Image and Text Generation" Fowl et al. (2020) is a paper that establishes a diversity metric that measures how wide a range of text or images a GAN is capable of outputting. The authors assert that there are many ways that GANs are being evaluated, but the diversity of their generation is often overlooked and that pre-existing metrics for measuring diversity in their generation were "rudimentary tools" which further emphasizes the importance of research on data quality.

