# OpenReview forum: "Data Alignment Predicts Language Model Performance: Evidence from Controlled Experiments in Autoformalization"
_ICLR.cc/2026/Conference — Submitted to ICLR 2026_

### Official Review · Reviewer_pEU9 · 2025-10-26

**Soundness:** 1
**Presentation:** 1
**Contribution:** 1
**Rating:** 0
**Confidence:** 3

**Summary:**

This paper examines the relationship between an LLM’s training and evaluating, claiming that data alignment (between training and evaluation) is a stronger predictor of performance than model or dataset size. The authors quantify “data alignment” using Task2Vec, an established tool from the meta-learning literature, and validate results with alternative metrics such as GZIP-Align and SBERT-Align.

**Strengths:**

-

**Weaknesses:**

- Limited novelty. The central claim that similarity between training and evaluation data improves downstream performance, is well-known in the machine learning community. The study primarily confirms this existing intuition.

- Short and underdeveloped. The manuscript reads more like an extended technical report than a complete conference paper. Besides, key components such as conclusion and related work are missing.

Given the limited novelty, incomplete structure, and short scope, I recommend rejection at this stage. If further investigation or follow-up analysis is pursued, I would be interested in contributing.

**Questions:**

I have no questions.

---

### Official Review · Reviewer_1Wu7 · 2025-10-28

**Soundness:** 1
**Presentation:** 2
**Contribution:** 1
**Rating:** 2
**Confidence:** 4

**Summary:**

This paper investigates whether data alignment is a stronger predictor of language model performance than dataset size. Through controlled experiments, the authors demonstrate that domain alignment between a train and test datasets consistently predicts low model perplexity.

**Strengths:**

* **Controlled setup**: The paper designs an end to end controlled setup isolating the impact of data and model perplexity.

**Weaknesses:**

* **Setup**:
  * The author used GPT-2 as the base architecture. This model was released in 2019, and many advancements have been made since then in regularization, normalization, and attention techniques, which have significantly improved the generalization of large language models.
  * The author assesses perplexity as a downstream evaluation. While perplexity is often correlated with downstream performance metrics (like F1 score, BERT score, or others), this is not always the case. Notably, in these experimentations, perplexity remains high across all different experimentations, which could mitigate its correlation with usual downstream performance metrics.
  * Following this observation, the models used during experimentation are smaller than standard small LMs (which habitually start at 500M parameters and above). However, the size of LLMs plays a crucial role in their modeling capacity, directly impacting perplexity values and generalization to out-of-domain training.
  * Training procedures are not clearly described. Due to the relatively small size of the fine-tuning datasets, with an average of only 4k tokens per dataset used, it seems necessary to provide details on the batch size, context length, and number of steps used.
Tokenizers play a crucial role in perplexity values. Training with different tokenizers seems necessary to consolidate the different claims.
* **Novelty**:
  * Data alignment is a well-known key component of model performance. This technique has been widely used in most recent models with multi-curriculum training (can also be referred to as an "annealing" phase), where data distribution is shifted to align with user requirements at the end of training to boost model performance. Data alignment is also the core idea behind model fine-tuning, where models are specialized on an in-domain distribution to boost model performance.
* **Cross capability**: While data alignment plays a crucial role in model performance, this is not always the case, as different works have shown that specific distributions can boost others. Notably, different multilingual works on cross-lingual representation, or recent studies on the impact of code and reasoning data, demonstrate enhancements in general model capabilities.

**Questions:**

* It could be interesting to use bigger language models, notably during fine-tuning. This could be achieved by performing out-of-domain fine-tuning starting from a pre-trained model with a known pre-training dataset to keep the experimental setup controlled. (cf. Pythia model family released a set of models with varying sizes and pre-training dataset mixes.)
* It would be valuable to complement perplexity with evaluations on downstream tasks, as perplexity does not always correlate with actual model performance, particularly given its strong dependence on tokenizer choices.
* Following the previous question, it would be valuable to use different tokenizers, as perplexity is highly correlated with tokenizer choice.

---

### Official Review · Reviewer_6KNv · 2025-10-31

**Soundness:** 2
**Presentation:** 1
**Contribution:** 1
**Rating:** 2
**Confidence:** 4

**Summary:**

This paper studies the degree to which the similarity between pretraining data and evaluation datasets can predict downstream performance. The paper measures similarity in multiple ways (Task2Vec embeddings, GZIP compression distance, sentence embeddings) and finds that all correlate with downstream performance. The paper pretrains GPT-2 models on three datasets and separately finetunes models on multiple datasets with an emphasis on autoformalization tasks for evaluation.

**Strengths:**

1. This paper's experiments identify a link between LLM performance and the similarity between training and evaluation data for 51M-parameter GPT-2 models. The paper studies both pretraining and finetuning scenarios.
2. The paper operationalizes similarity between the training data and evaluation data in multiple ways, with similar findings across each.

**Weaknesses:**

1. The pretraining setups are limited and small. The 51M-parameter GPT-2 models were trained for 1.31B tokens. While it is always easy to ask for larger experiments, this setup is very far from current practice and current analysis. The takeaways would be stronger if they hold with larger models and larger training datasets.

2. The pretraining datasets (PubMed abstracts, USPTO backgrounds, and a combination of both) are much simpler than the very heterogeneous pretraining datasets typically in use today. Do the findings generalize from this very targeted setup to the standard way LLMs are trained on more diverse data? Existing work suggests that standard LLM training setups may be more challenging [1 (Figure 5), 2].

3. The paper does not sufficiently discuss prior work on the topic. How does this paper go beyond, for example, paper [1] below, which studies the degree to which similarity between pretraining data and downstream examples (as measured with kernel density estimation) predicts performance?

    In a similar vein, lines 39-40 state that "Current discourse predominantly highlights the scale of a dataset as a pivotal factor in its capacity to effectively pre-train or fine-tune a model". While I agree that scale is often the most-discussed aspect of data, there is much related work that studies the degree to which training data quality (not just scale) can impact performance [3, 4, 5, 6].

3. The motivation for studying autoformalization is not clear to me. The paper would be made stronger by a) explaining in the introduction why study autoformalization in particular and b) studying alignment for a wider variety of downstream tasks when finetuning in Section 3.4

4. Additional finetuning experiments with larger and more varied finetuning datasets would lend additional evidence to this part of the paper. The existing finetuning experiments use two small datasets of 1.4k tokens and 4.1k tokens.

[1]: LMD3: Language Model Data Density Dependence, Kirchenbauer et al., COLM 2024.

[2]: Data Similarity is Not Enough to Explain Language Model Performance, Yauney et al., EMNLP 2023.

[3]: Dolma: an Open Corpus of Three Trillion Tokens for Language Model Pretraining Research, Soldaini et al., ACL 2024.

[4]: The FineWeb Datasets: Decanting the Web for the Finest Text Data at Scale, Penedo et al, 2024.

[5]: A Pretrainer’s Guide to Training Data: Measuring the Effects of Data Age, Domain Coverage, Quality, & Toxicity, Longpre et al., NAACL 2024.

[6]: Scaling Laws for Data Filtering -- Data Curation cannot be Compute Agnostic, Goyal et al., CVPR 2024.

**Questions:**

1. Are the reported $r^2$ correlation values statistically significant?

Small comments:
- Line 92 -> \citet{} -> \citep{}
- Table 1: I suggest using the booktabs package to make the table look less cluttered, with fewer dividing lines.
- Figure 3 would be better placed in an appendix rather than the main paper.
- The style of Related Work (Section F.2) is nonstandard. Papers should be referenced with \citet{} in text rather than writing out the titles.

---

### Official Review · Reviewer_XfgN · 2025-10-31

**Soundness:** 1
**Presentation:** 1
**Contribution:** 1
**Rating:** 0
**Confidence:** 5

**Summary:**

The paper examines the relationship between train set / test set alignment and model performance, with very limited experiments in LLM pre-trained and fine-tuning.

**Strengths:**

The writing is, for the most part, clear.

**Weaknesses:**

The paper is clearly unfinished and in my opinion therefore not fit for review.

**Questions:**

N/A.

**Details Of Ethics Concerns:**

The paper is clearly unfinished.

---

### Meta-Review · Area_Chair_d7GC · 2026-01-07

**Summary:**

This paper demonstrates that the alignment coefficient between pre-training/fine-tuning data and the evaluation data predicts downstream performance on evaluation data. They initially show the results using Task2Vec embeddings to compute an alignment coefficient and then verify this with GZIP-Align (compression distance) and SBERT-Align (embedding cosine). The reviewers consider as strength the controlled setup of the paper and the observed link between LLM performance and the similarity between training and evaluation data.

**Reviewer Concerns:**

Reviewer pEU9:
- **Limited novelty**. The central claim that similarity between training and evaluation data improves downstream performance, is well-known in the machine learning community.
- **Short and underdeveloped**

Reviewer 1Wu7:
- **Recent architecture should be evaluated**: The author used GPT-2 (2019) as the base architecture.
- **Perplexity as evaluation metric**: While perplexity is often correlated with downstream performance metrics (like F1 score, BERT score, or others), this is not always the case.
- **Small models**
- **Missing experimental details**
- **Data alignment is well-known**

Reviewer 6KNv:
- **Limited and small setups**
- **Simple pretraining datasets**
- **Missing prior work discussion**
- **Missing motivation for studying autoformalization**

**Reviewer Scores:**

Reviewers gave scores 0, 0, 2, 2. The authors did not respond during the rebuttal. The reviewers noted various limitations including a small scale of experimental setup, limited novelty, and underdeveloped writing.

---

### Decision · Program_Chairs · 2026-01-26

Reject